# A Parameter Self-Tuning Decoupling Controller Based on an Improved ADRC for Tension Systems

**Guoli Ju** [1,†]**, Shanhui Liu** [1,*]**, Keliang Wei** [2]**, Haodi Ding** [1] **and Chaoyue Wang** [1]

1   Faculty of Printing, Packaging Engineering and Digital Media Technology, Xi'an University of Technology, Xi'an 710048, China; 2220821107@stu.xaut.edu.cn (G.J.); 2210820061@stu.xaut.edu.cn (H.D.); 2220821141@stu.xaut.edu.cn (C.W.)
2   Shaanxi Beiren Printing Machinery Co., Ltd., Weinan 714000, China; weikeliang@shaanxibeiren.com
*   Correspondence: shanhuiliu@xaut.edu.cn; Tel.: +86-1592-972-4880

**Abstract:** Aiming at the problems of strong coupling and time-varying parameters in the tension systems of roll-to-roll coating machines, this paper outlines the design of a parameter self-tuning decoupling controller based on an improved active disturbance rejection controller (ADRC) for tension systems. First, we established a nonlinear coupled model for a global tension system based on the roll-to-roll coating machine tension constituent unit and working principle. Second, we introduced virtual quantities and actual control quantities; then, we applied an ADRC for decoupling control of the tension system based on a nonlinear coupling model, using a genetic algorithm (GA) to achieve parameters tuning in the ADRC. Finally, both the traditional proportional–integral–differential (PID) controller and the designed controller were simulated to evaluate their anti-interference and decoupling performance. Furthermore, the performances of several controllers were compared to the results obtained by other researchers. The result found that the parameter self-tuning decoupling controller based on an improved ADRC for tension systems exhibited better decoupling control performance and that the controller was able to achieve precise control of tension.

**Keywords:** roll-to-roll coating machine; tension system; decoupled control; ADRC; parameter self-tuning





## 1. Introduction

A roll-to-roll coating machine is an excellent option for producing printed electronic devices in a green, efficient, and continuous manner due to its thick coating and stable process. For the manufacturing of flexible electronic devices, it is crucial to maintain steady-state tension, with substrate tension control accuracy being required to be within ±5%. The accuracy of substrate tension control affects the quality of flexible electronic devices, causing defects such as folds, collapse, and slip. Therefore, tension control accuracy is critical for substrates. However, due to the tension system having multiple inputs and outputs, strong coupling, and parameters that vary with time, precise tension control is challenging.

When it comes to modeling the tension systems of roll-to-roll coating machines, scholars take into account factors such as motor speed difference, bearing misalignment, shaft roundness error, and the impact of drying temperature on the substrate's elastic modulus. They then study and establish a serial coupling model of the tension system based on these factors. At the same time, many scholars have proposed new control strategies to improve tension control accuracy in roll-to-roll coating manufacturing in response to increasing accuracy requirements. Brandenburg et al. first proposed a multi-span tension model based on the law of mass conservation [1]. Lee et al. introduced thermal deformation and equivalent elastic modulus into the multi-span tension model, and they researched and analyzed the changes these two factors made to the substrate tension [2]. Jaehyeong et al. analyzed the factors that contribute to the variation of

tension in a roll-to-roll system and developed a tension model for each section, which successfully predicted the tension applied to the system [3]. Lu et al. established a heat transfer model capable of predicting the temperature distribution of the substrate [4]. Ying et al., based on the energy approach and the heat conduction equation, considering the effect of deformation, derived the nonlinear vibrational equations of an axially traveling flexible printing electronic membrane coupled with temperature under the function of a nonlinear electrostatic excitation force [5]. Lee et al. established a mathematical model to express the transverse motion of the multi-span substrate [6]. Wu et al. trained a special neural network model to dynamically update rewinding roll tension with the radius of the roll [7]. Jin et al. proposed a model for estimating web tension and tension distribution by measuring the local contact force on the substrate [8]. Research on tension control strategy has also been greatly developed. Based on research on the modeling of the tension system, Liang et al. proposed a coordinated and optimized control method for multiple rolls, in which the speed of the drive roll of the next unit was adjusted to maintain the low tension condition of the substrate when the speed of the drive roll of the previous unit changes [9]. Chen et al. proposed a nonlinear model predictive control scheme to eliminate tension perturbations in roll-to-roll systems [10]. Jeon et al. proposed a feed-forward controller that employed finite element method (FEM)-based web temperature distribution inputs to enhance tension control performance during the drying span [11]. Liu et al. proposed a parameter self-tuning and self-anti-turbulence control strategy based on an RBF neural network by integrating feed-forward, RBF, and an ADRC, which improved the accuracy of the tension control [12]. Ding et al. proposed a feed-forward improved PID parameter self-tuning and decoupling controller, which realized the steady-state control of tension [13]. Kim et al. proposed a perturbed machine model that made it possible to handle both nonlinearities and parameter variations of a machine using simple static and first-order dynamic compensators [14]. Hwang et al. fused a disturbance observer, a feed-forward controller, and a Kalman filter to realize the observation of model disturbance, signal tracking, and signal processing, which greatly improved the performance of tension control [15].

This paper proposes a parameter self-tuning decoupling controller based on an improved ADRC for tension systems in roll-to-roll coating machines. Section 2 analyzes a machine's tension system structure and establishes a global coupling model of the tension system. Section 3 outlines the design of the parameter self-tuning decoupling controller based on an improved ADRC for tension systems, and Section 4 describes the simulation and verifies its performance combined with the global coupling model. Section 5 summarizes this paper.

## 2. Modeling and Analysis of Tension Systems for Roll-to-Roll Coating Machines

Figure 1 illustrates the structure of the tension system of a roll-to-roll coating machine, which mainly consists of three major subsystems: the unwinding tension subsystem, the coating tension subsystem, and the rewinding tension subsystem. Among these, the unwinding tension subsystem consists of an unwinding and an unwinding infeed unit, the coating tension subsystem consists of a coating unit, and the rewinding tension subsystem consists of a rewinding and a rewinding outfeed unit. Additionally, there is the dancer roll mechanism set up in the unwinding, rewinding outfeed, and the rewinding unit, the tension sensor set up in the unwinding outfeed and the coating unit, and an oven set up in the coating unit.

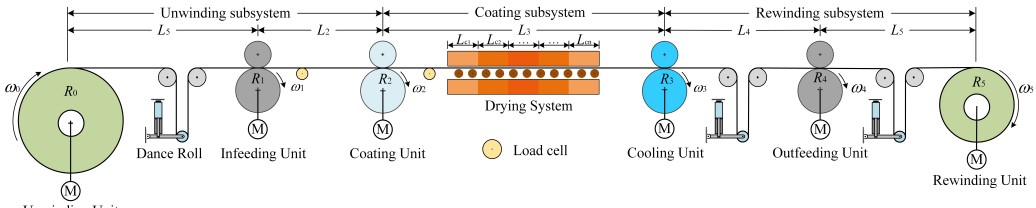

**Figure 1.** The roll-to-roll coating machine structure of the tension system.

According to the literature [16], combined with the specific structure of the tension system in Figure 1, the global tension system coupling model of the roll-to-roll coater is established as follows in Equation (1), and the meanings of the parameters in Equation (1) are shown in Table 1.

$$
\left\{
\begin{aligned}
&\begin{bmatrix} \frac{dT_1(t)}{dt} \\ \frac{dT_2(t)}{dt} \\ \frac{dT_3(t)}{dt} \\ \frac{dT_4(t)}{dt} \\ \frac{dT_5(t)}{dt} \end{bmatrix} =
\begin{bmatrix} \frac{[AE-T_1(t)]}{L_1(t)}\frac{dL_1(t)}{dt} \\ 0 \\ \frac{[AE-T_2(t)]}{L_3} \\ \frac{[AE-T_4(t)]}{L_4(t)}\frac{dL_4(t)}{dt} - \frac{[AE-T_2(t)]R_3\omega_3}{L_4(t)} \\ \frac{[AE-T_5(t)]}{L_5(t)}\frac{dL_5(t)}{dt} \end{bmatrix} + \\
&\begin{bmatrix}
-\frac{[AE-T_s(t)]R_0}{L_1(t)} & \frac{[AE-T_1(t)]R_1}{L_1(t)} & 0 & 0 & 0 \\
0 & -\frac{[AE-T_1(t)]R_1}{L_2} & \frac{[AE-T_2(t)]R_2}{L_2} & 0 & 0 \\
0 & 0 & -\frac{[AE-T_2(t)]R_2}{L_3} & 0 & 0 \\
0 & 0 & 0 & \frac{[AE-T_4(t)]R_4}{L_4(t)} & 0 \\
0 & 0 & 0 & -\frac{[AE-T_4(t)]R_4}{L_5(t)} & \frac{[AE-T_5(t)]R_5(t)}{L_5(t)}
\end{bmatrix}
\begin{bmatrix} \omega_0(t) \\ \omega_1(t) \\ \omega_2(t) \\ \omega_4(t) \\ \omega_5(t) \end{bmatrix} \\
&L_3 = \left[ L_3^* + (L_{c1}-1)\frac{E}{E_{c1}} + (L_{c2}-1)\frac{E}{E_{c2}} + \cdots + (L_{cn}-1)\frac{E}{E_{cn}} \right]
\end{aligned}
\right. \tag{1}
$$

**Table 1.** Meaning of coupled model parameters.

| Parameters | Meaning |
|---|---|
| $T_1 \sim T_5$ | The tension of the substrate in each unit |
| $L_1 \sim L_5$ | The length of the substrate in each unit |
| $\omega_0 \sim \omega_2$ | The angular speed of each motor |
| $R_0 \sim R_5$ | The radius of the unwinding, drive rolls, and rewinding |
| $L_{c1} \sim L_{cn}$ | Length of each section of the drying mechanism |
| $E_{c1} \sim E_{cn}$ | Young's modulus of elasticity of substrate at different temperatures |
| $L_3^*$ | Length of the substrate without drying mechanism |
| $A$ | The cross-sectional area of the substrate |
| $E$ | Young's modulus of elasticity of substrate |

The sensor should be placed close to the driving roll that controls the substrate tension in the unwinding infeed and coating unit. This ensures accurate measurement of tension [17]. The dancer roll mechanism is located in the unwinding, rewinding outfeed, and rewinding unit. It consists of the dancer roll, dancer rod, and cylinder that provides thrust to the dancer roll [18] and is shown in Figure 2. When the cylinder thrust torque and the substrate tension torque are in equilibrium, the dancer roll remains in a stable position. However, when the substrate tension changes, the dancer roll deflects and produces a deflection angle of $\theta$. The substrate tension can be indirectly calculated through the dancer roll mechanism tension detection model.

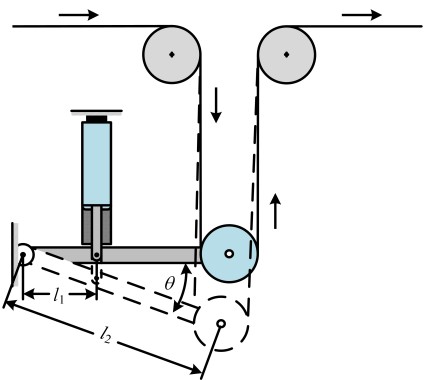

**Figure 2.** The structure of dancer roll mechanism.

In addition, when the dancer roll mechanism deflects, the lengths of $L_1$, $L_4$, and $L_5$ will also change. Similarly, changes in the size of $R_0$ and $R_5$ during coating manufacturing will also impact the lengths of $L_1$ and $L_5$. Based on literature [16] and the specific structure of the tension system in Figure 1, establish the tension detection model of the dance roll and the substrate length change model as follows in Equation (2), the meaning of the parameters in Equation (2) are shown in Table 2.

$$
\begin{cases}
T_1(t) = \frac{1}{2l_2}\left\{-J_\mathrm{D}\theta_1'' - B\theta_1' + l_1[F_\mathrm{P} - kl_1\theta_1(t)]\right\} \\
L_1(t) = L_1^* + \sqrt{L_{d1} - [R_0(t) - r_1]^2} - 2l_2\theta_1 \\
T_4(t) = \frac{1}{2l_2}\left\{-J_\mathrm{D}\theta_4'' - B\theta_4' + l_1[F_\mathrm{P} - kl_1\theta_4(t)]\right\} \\
L_4(t) = L_4^* - 2l_2\theta_4 \\
T_5(t) = \frac{1}{2l_2}\left\{-J_\mathrm{D}\theta_5'' - B\theta_5' + l_1[F_\mathrm{P} - kl_1\theta_5(t)]\right\} \\
L_5(t) = L_5^* + \sqrt{L_{d2} - [R_5(t) - r_2]^2} - 2l_2\theta_5
\end{cases}
\tag{2}
$$

**Table 2.** Meaning of the parameters of the auxiliary equation of tension.

| Parameters | Meaning |
| --- | --- |
| $J_D$ | The equivalent rotational inertia of the pendulum arm |
| $F_P$ | The cylinder thrust |
| $B$ | The friction coefficient |
| $r_1, r_2$ | The size of the radius of the guiding roller in the dancer roll mechanism |
| $L_1^*$ | The length of substrate in the unwinding unit in the initial steady state |
| $L_5^*$ | The length of substrate in the rewinding unit in the initial steady state |
| $l_1$ | Distance between the center of rotation of the dance roll and the connection point |
| $l_2$ | The length of the pendulum rod |
| $k$ | Spring force coefficient of the spring in the cylinder |

## 3. Decoupled Controller Design for Global Tension System

### 3.1. Tension System Coupled Model Decoupling Control

The tension system global coupling model of a roll-to-roll coating machine is a typical multi-input and output system. To transform it into a single-input single-output system, virtual control quantities $u(t)$ are introduced, followed by the use of the static decoupling model to determine the actual control quantities $\omega(t)$, resulting in the decoupling control of the tension system. Equation (1) is written in the form of a matrix:

$$
\frac{\mathrm{d}\boldsymbol{T}(t)}{\mathrm{d}t} = \boldsymbol{f}(t) + \boldsymbol{B}(t)\boldsymbol{\omega}(t)
\tag{3}
$$

Among them:

$$\boldsymbol{T}(t) = \begin{bmatrix} T_1(t) & T_2(t) & T_3(t) & T_4(t) & T_5(t) \end{bmatrix}^T \tag{4}$$

$$\boldsymbol{\omega}(t) = \begin{bmatrix} \omega_0(t) & \omega_1(t) & \omega_2(t) & \omega_4(t) & \omega_5(t) \end{bmatrix}^T \tag{5}$$

$$f(t) = \begin{bmatrix} f_1(t) \\ f_2(t) \\ f_3(t) \\ f_4(t) \\ f_5(t) \end{bmatrix} = \begin{bmatrix} \frac{[AE-T_1(t)]}{L_1(t)}\frac{\mathrm{d}L_1(t)}{\mathrm{d}t} \\ 0 \\ \frac{[AE-T_2(t)]}{L_3} \\ \frac{[AE-T_4(t)]}{L_4(t)}\frac{\mathrm{d}L_4(t)}{\mathrm{d}t} - \frac{[AE-T_2(t)]R_3\omega_3}{L_4(t)} \\ \frac{[AE-T_5(t)]}{L_5(t)}\frac{\mathrm{d}L_5(t)}{\mathrm{d}t} \end{bmatrix} \tag{6}$$

$$\boldsymbol{B}(t) = \begin{bmatrix} -\frac{[AE-T_s(t)]R_0}{L_1(t)} & \frac{[AE-T_1(t)]R_1}{L_1(t)} & 0 & 0 & 0 \\ 0 & -\frac{[AE-T_1(t)]R_1}{L_2} & \frac{[AE-T_2(t)]R_2}{L_2} & 0 & 0 \\ 0 & 0 & -\frac{[AE-T_2(t)]R_2}{L_3} & 0 & 0 \\ 0 & 0 & 0 & \frac{[AE-T_4(t)]R_4}{L_4(t)} & 0 \\ 0 & 0 & 0 & -\frac{[AE-T_4(t)]R_4}{L_5(t)} & \frac{[AE-T_5(t)]R_5(t)}{L_5(t)} \end{bmatrix} \tag{7}$$

$\boldsymbol{T}(t)$, $\boldsymbol{\omega}(t)$ is the tension system output and input, $\boldsymbol{f}(t)$ is the tension system control quantities of the outer part, known as the "dynamic coupling" part; $\boldsymbol{B}(t)\boldsymbol{\omega}(t)$ is the "static coupling" part of the system. Introducing the virtual control quantities $\boldsymbol{u}(t) = [u_1(t)u_2(t)u_3(t)u_4(t)u_5(t)]^T$, so that $\boldsymbol{u}(t) = \boldsymbol{B}(t)\boldsymbol{\omega}(t)$, then Equation (3) becomes:

$$\frac{\mathrm{d}\boldsymbol{T}(t)}{\mathrm{d}t} = \boldsymbol{f}(t) + \boldsymbol{u}(t) \tag{8}$$

From Equation (8), $\boldsymbol{u}(t)$ and $\mathrm{d}\boldsymbol{T}(t)/\mathrm{d}t$ form a single-input single-output system to realize the tension system decoupling. $\boldsymbol{f}(t)$ is the sum of perturbations of each unit when applying the ADCR controller to control substrate tension, the disturbance sum $\boldsymbol{f}(t)$ can be estimated and compensated, while the virtual control quantities $\boldsymbol{u}(t)$ can be obtained. Then, the actual input quantities $\boldsymbol{\omega}(t)$ can be obtained using Equation (9), which realizes decoupling control of the tension system.

$$\boldsymbol{\omega}(t) = \boldsymbol{B}^{-1}(t)\boldsymbol{u}(t) \tag{9}$$

$$|\boldsymbol{B}(t)| = -\frac{[AE-T_s(t)][AE-T_1(t)][AE-T_3(t)][AE-T_4(t)][AE-T_5(t)]R_0(t)R_1R_2R_4R_5(t)}{L_1(t)L_2L_kL_4(t)L_5(t)} \tag{10}$$

$$\boldsymbol{B}^{-1}(t) = \begin{bmatrix} -\frac{[AE-T_s(t)]R_0}{L_1(t)} & \frac{[AE-T_1(t)]R_1}{L_1(t)} & 0 & 0 & 0 \\ 0 & -\frac{[AE-T_1(t)]R_1}{L_2} & \frac{[AE-T_2(t)]R_2}{L_2} & 0 & 0 \\ 0 & 0 & -\frac{[AE-T_2(t)]R_2}{L_3} & 0 & 0 \\ 0 & 0 & 0 & \frac{[AE-T_4(t)]R_4}{L_4(t)} & 0 \\ 0 & 0 & 0 & -\frac{[AE-T_4(t)]R_4}{L_5(t)} & \frac{[AE-T_5(t)]R_5(t)}{L_5(t)} \end{bmatrix} \tag{11}$$

Since the product of the substrate cross-sectional area $A$ and Young's modulus of elasticity $E$ is much larger than the substrate tension in each cell, $|\boldsymbol{B}(t)| \neq 0$, $\boldsymbol{B}^{-1}(t)$ exists, and Equation (11) is called the static decoupling model.

### 3.2. GA-Based Self-Tuning of ADRC Parameters

The better robustness of the active disturbance rejection controller(ADRC) compared to the conventional proportional-integral-differential(PID) controller makes it a good choice for applicators, and current applications range from power electronics, motion control, and tension control [19]. The global tension system coupling model uses five ADRCs with the same structure for tension control. This ADRC consists of four main components, namely the most rapid tracking differentiator(TD), the first-order nonlinear error feedback control(NLESF), the first-order dilated state observer(ESO), and compensating the system based on the perturbation estimation, with the discrete formulas as shown below:

$$
\begin{cases}
e_T(k) = x_1(k) - T_0(k) \\
fh(k) = \text{fhan}(e_T(k), x_2(k), r_0, h_0) \\
x_1(k+1) = x_1(k) + hx_2(k) \\
x_2(k+1) = x_2(k) + hfh(k) \\
e_E(k) = z_1(k) - T(k) \\
z_1(k+1) = z_1(k) + h[z_2(k) - \beta_1 e_E(k) + b_0 u(k)] \\
z_2(k+1) = z_2(k) + h[-\beta_2 fal(e_E(k), \alpha_1, \delta_1)] \\
e_N(k+1) = x_1(k+1) - z_1(k+1) \\
u(k+1) = K_p fal(e_N(k+1), \alpha_2, \delta_2) - z_2(k+1)/b_0
\end{cases}
\tag{12}
$$

where $r_0$ is the time scale that responds to the response speed of the system, $h_0$ is the filtering factor that determines the effect of filtering against noise [20]; and $\alpha_1$ and $\alpha_2$ are the nonlinear factors [21], which are quantities reflecting the speed of convergence of the function, $0 < \alpha < 1$; $\delta_1, \delta_2$ are the filtering factors [21], which are quantities that determine the length of the linear segment of the function; $b_0$ is a quantity related to the system itself, which is a quantity that responds to the strength of the compensation of the system error; $\beta_1, \beta_2$, and $K_p$ are the quantities of the function gain in ESO and NLESF; $h$, on the other hand, is related to the simulation accuracy.

The above 11 parameters, except for the three parameters of $\beta_1, \beta_2$, and $K_p$, can be selected empirically, and Table 3 shows the parametric quantities selected empirically, while the three parameters of $\beta_1, \beta_2$, and $K_p$ are directly related to the control effect due to the intrinsic connection between each other, and are directly related to the control effect. The traditional empirical selection is characterized by trial and error and has certain limitations. Genetic algorithm (GA) has been successfully applied to a wide range of problems due to its simplicity, global perspective, and intrinsic parallel processing [22]. Therefore, the optimization tuning of the three parameters is carried out using GA, and the parametric quantities that rely on the empirical method of initialization are shown in Table 4.

**Table 3.** Empirical Selection of Parameters.

| Parameters | $r_0$ | $h_0$ | $\alpha_1$ | $\alpha_2$ | $\delta_1$ | $\delta_2$ | $b_0$ | $h$ |
|---|---|---|---|---|---|---|---|---|
| **Value** | 600 | 0.6 | 0.5 | 0.5 | 0.01 | 0.01 | 1 | 0.01 |

**Table 4.** GAoptimization parameters.

| Parameters | $K_p$ | $\beta_1$ | $\beta_2$ |
|---|---|---|---|
| **Value** | 100 | 50 | 6500 |

GA typically begins its search with a set of random solutions encoded in a binary string structure or floating-point numbers. Each solution is assigned a value related to the objective function. The population of solutions is then modified into a new population by applying three operators similar to natural genetic operators—reproduction, crossover, and mutation. The GA works iteratively by applying these three operators successively in each

generation until the termination criterion is satisfied [22]. The specific setup of the GA in this paper is as follows:

1. Population initialization: Population range, number, number of iterations, and coding method are shown in Table 5.

**Table 5.** Table of initialization settings for populations.

| Parameters | | | Population | Number of Iterations | Coding Method |
|---|---|---|---|---|---|
| $K_p$ | $\beta_1$ | $\beta_2$ | | | |
| $50 \sim 1000$ | $50 \sim 2000$ | $10^3 \sim 10^7$ | 100 | 100 | Floating-point |

2. Population update: Specific choices for the three operations performed by the population update are shown in Table 6.

**Table 6.** Table of population renewal settings.

| Way | Elite Choice | Hybrid crossover | Gaussian approximate mutation |
|---|---|---|---|
| **Probability** | 0.01 | 0.2 | 0.3 |

3. Population fitness evaluation: For the tension control problem, the GA objective is to optimize tuning the three parameters of $\beta_1$, $\beta_2$, and $K_p$ in the ADRC to control the tension error to the minimum—minimization problem [22], so the inverse of the objective function is selected as the fitness function, and the objective function is selected as the integral of absolute error (IAE) [20], taking into account the characteristics of the steady-state control of tension, the objective function is selected based on IAE, the introduction of tension overshooting (Mp) and the IAE together with the objective function, and Equation (13) expresses the selected fitness function and the objective function,$w_1$, $w_1$ are the weighting parameters.

$$\begin{cases} f(x_1, x_2, \ldots, x_n) = \frac{1}{1 + J(x_1, x_2, \ldots, x_n)} \\ J(x_1, x_2, \ldots, x_n) = \int_0^\infty \left[ w_1|e(t)| + w_2 M_p(t) \right] dt \end{cases} \tag{13}$$

The Equation (13) provides a discrete expression for the tension error $e(t)$ is given below:

$$\begin{cases} e(k) = z_1(k) - T_s(k) \\ z_1(k+1) = z_1(k) + h[z_2(k) - \beta_1 e_E(k) + b_0 u(k)] \\ z_2(k+1) = z_2(k) + h[-\beta_2 e_E(k)] \\ u(k+1) = K_P e_L(k+1) - z_2(k+1)/b_0 \end{cases} \tag{14}$$

The parameter self-tuning decoupling controller based on improved ADRC for the tension system is designed based on the static decoupling of the global coupling model of the tension system and parameter tuning of the ADRC using GA. The controller has two phases of tension control that follow the same process. In phase one, the tension error $e(t)$ is judged every 1 s, starting from 0.5 s. If $e(t)$ is greater than 0.5 at the detection moment, the parameters of $\beta_{1i}$, $\beta_{2i}$, and $K_{ip}$ are optimized and adjusted using GA, and the parameters remain unchanged in the opposite way. In phase two, the ADRC controllers use the three parameters optimized tuning in phase one and output the virtual control quantity $\boldsymbol{u}(t)$ and then output the actual control quantity $\boldsymbol{\omega}(t)$ after the static decoupling model. The $\boldsymbol{\omega}(t)$ is then input into the global tension system to achieve steady-state control of the global tension system. The schematic representation of the controller is shown in Figure 3, and the working principle of this controller is shown in Figure 4.

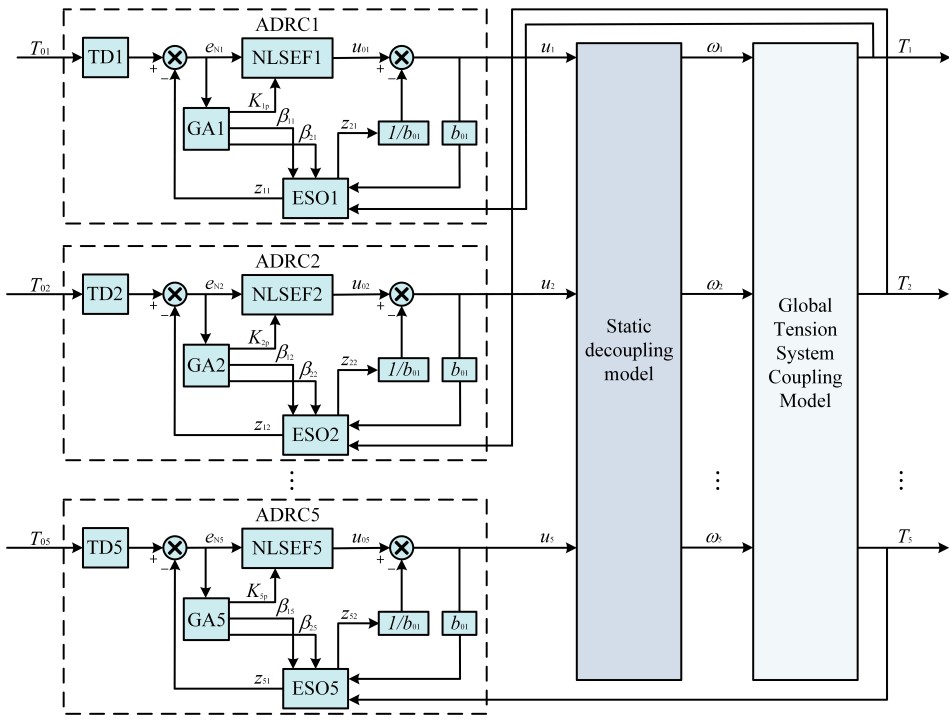

**Figure 3.** The global tension system decoupling controller.

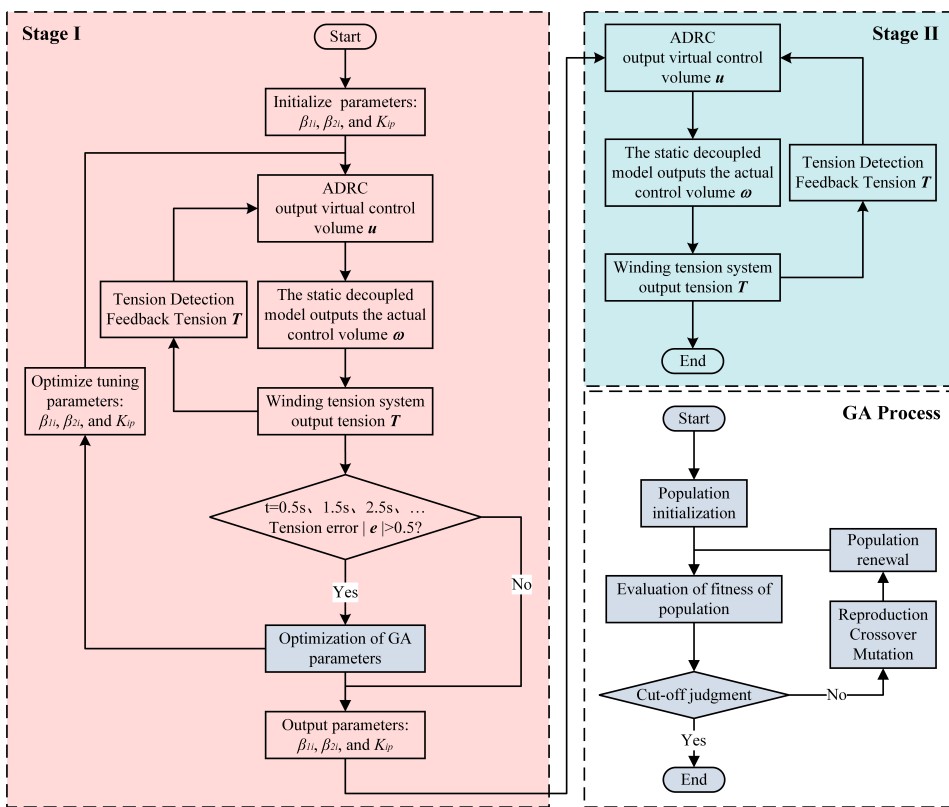

**Figure 4.** Global tension system decoupling controller tension control flow chart.

## 4. Simulation and Analysis of Decoupled Controller for Global Tension System

To test the performance of the proposed self-tuning decoupling controller based on improved ADRC for tension system, a global tension system model is created in Simulink (MATLAB2019a). This model is then used to simulate and analyze the proposed decoupling controller to verify its parameters tuning effect using GA-based ADRC. The simulation also

compares the proposed decoupling controller with the traditional PID controller in terms of decoupling performance and anti-interference performance. This helps to analyze the decoupling control effect of the proposed decoupling controller. The simulation is carried out with a synchronized line speed of 300 m/min and one section-type oven set up in the coating unit. The parameters of the global coupling model of the tension system and each PID controller are shown in Tables 7 and 8, respectively.

**Table 7.** Global tension system parameters.

| Parameters | Value | Units |
|---|---|---|
| $A$ | $2.7 \times 10^{-5}$ | $m^2$ |
| $E$ | $4.89 \times 10^9$ | Pa |
| $E_{c1}$ | $0.463 \times 10^9$ | Pa |
| $R_0$ | 0.5 | m |
| $R_1, R_4$ | 0.0925 | m |
| $R_2$ | 0.93 | m |
| $R_3$ | 0.3 | m |
| $R_5$ | 0.5 | m |
| $r_1, r_2$ | 0.152 | m |
| $L_1^*$ | 4.628 | m |
| $L_2$ | 2.65 | m |
| $L_3$ | 20.2 | m |
| $L_4^*$ | 2.3 | m |
| $L_5^*$ | 1.391 | m |
| $L_{d1}, L_{d2}$ | 10 | m |
| $L_{c1}$ | 3.5 | m |
| $l_1$ | 0.154 | m |
| $l_2$ | 0.35 | m |

**Table 8.** The parameters list of PID controller

| Controller | Parameters | | |
|---|---|---|---|
| | $k_p$ | $k_i$ | $k_d$ |
| PID1 | 1 | 12 | 0 |
| PID2 | 1 | 12 | 0 |
| PID3 | 1 | 12 | 0 |
| PID4 | 1 | 12 | 0 |
| PID5 | 1 | 12 | 0 |

*4.1. Simulation and Analysis of Self-Tuning of ADRC Parameters Based on GA*

Set the simulation time to 10 s, optimize tuning $\beta_1$, $\beta_2$, $K_p$ in each ADRC based on the steps in Figure 4, and the parameter change curves during the simulation are as follows:

Figure 5 shows GA is used to tune initialization parameters at a specific time point. The results in Figure 5a–c demonstrate that all three parameters of the ADRCs are tuned at only 0.5 s, except for the three-parameter values of the ADRCs in Group 3, which fail to meet the requirements for adjusting parameters and, therefore, do not change. Figure 6 demonstrates the results of the simulation using initialization and optimized tuning parameters. Simulation results from (a), (b), (d), and (e) indicate that the tension of each segment has different degrees of oscillation and overshooting at the beginning and finally tends to be stable when the initialization parameters are used for simulation, while the tension of each segment is stable within the set range with no oscillations or overshooting when using optimization adjust parameters. Results (c) are the same because ADRC3's initialization parameters are optimized for tuning.

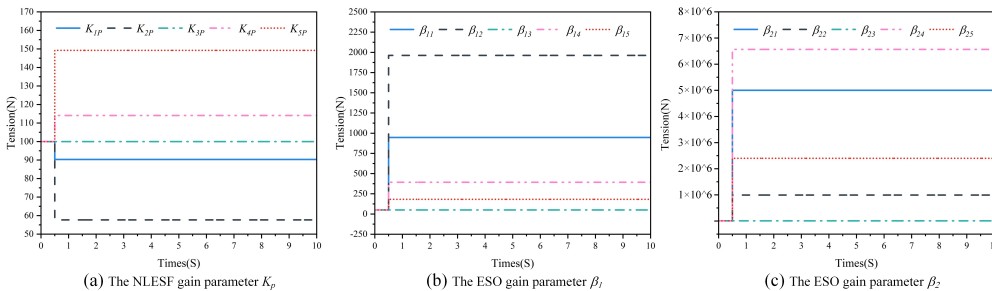

**Figure 5.** Parameter tuning result graph.

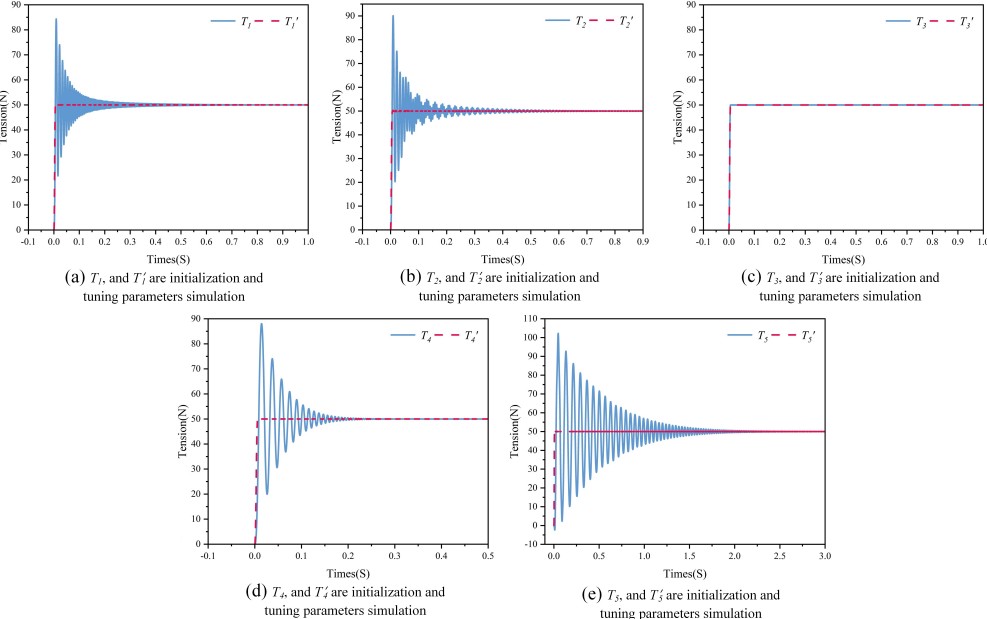

**Figure 6.** Comparison of simulation results of initialization and self-tuning parameters.

Based on the comparison, it can be concluded that the tuning of ADRC parameters using GA is more efficient and effective compared to the traditional empirical method of parameter tuning. GA is a good solution for the problem of lengthy and complicated tuning in ADRC. Table 9 displays the results of optimized tuning of the parameters in Table 4 using GA.

**Table 9.** Table of optimized tuning parameters.

| Controller | Parameters | | |
|---|---|---|---|
| | $K_p$ | $\beta_1$ | $\beta_2$ |
| ADRC1 | 90.39 | 947.91 | $5.00 \times 10^6$ |
| ADRC2 | 57.67 | 1962.04 | $9.9248 \times 10^5$ |
| ADRC3 | 100 | 50 | 6500 |
| ADRC4 | 114.07 | 392.92 | $6.5642 \times 10^6$ |
| ADRC5 | 149.23 | 181.50 | $2.3986 \times 10^6$ |

### 4.2. Simulation and Analysis of Decoupled Controller for Global Tension System

After optimally tuning the ADRC parameters using GA, each controller uses the optimally tuned parameters in Table 7 to conduct simulation experiments on the designed parameter self-tuning decoupling controllers based on improved ADRC for tension system in terms of decoupling performance and anti-interference capability and to compare with traditional PID controllers to verify its control accuracy.

### 4.2.1. Decoupling Performance Analysis

The system is in a steady state, and at the 2 s moment, $T_{02}$ is made to produce a tension step of duration 2 s and size 5 N; at the 6 s moment, $T_{04}$ is made to produce a tension step of duration 2 s and size $-5$ N. The results of the two experiments simulated using the parameter self-tuning decoupling controller based on improved ADRC for tension system and PID controller are shown in Figure 7.

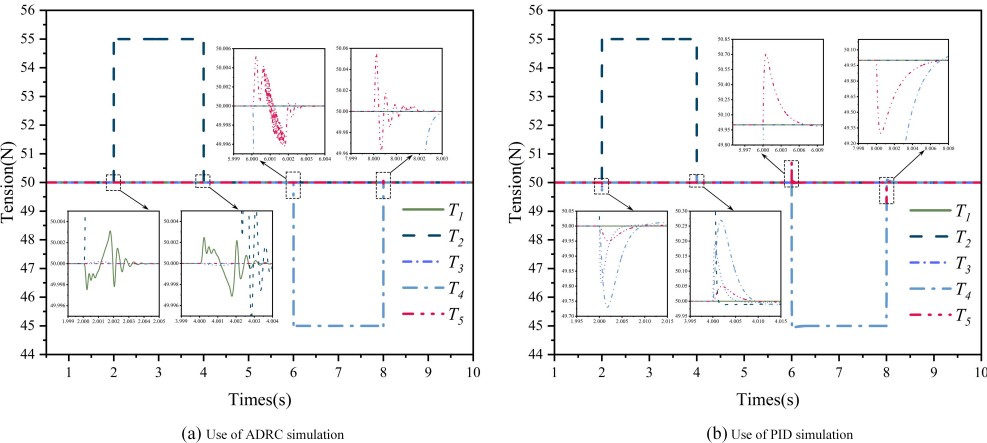

(a) Use of ADRC simulation          (b) Use of PID simulation

**Figure 7.** Comparison of decoupling performance simulation results.

The graphs in Group (a) of Figure 7 show that when $T_{02}$ generates a tension step, $T_1$ produces a 0.01% tension disturbance at around 2 s and 4 s. Similarly, when $T_{04}$ generates a tension step, $T_5$ produces a 0.012% tension disturbance near the moments of 6 s and 8 s. On the other hand, the graphs in Group (b) indicate that when $T_{02}$ produces a tension step, $T_3$, $T_4$, and $T_5$ produce tension perturbations of 0.4%, 0.6%, and 0.1% at the 2 s moment, and 0.4%, 0.6%, and 0.1% at the 4 s moment; when $T_{04}$ produces a tension step, $T_5$ produces tension perturbations of 1.4% and 1.5% at the 6 s and 8 s moments.

The above results show that the maximum relative error of the tension unit is 0.012% when using the parameter self-tuning decoupling controller based on improved ADRC for the tension system, while the maximum relative error of the tension unit is 1.5% when using the traditional PID controller for tension control. Compared with the traditional PID control, the parameter self-tuning decoupling controller based on improved ADRC for tension system has a better decoupling effect.

### 4.2.2. Anti-Interference Capability Analysis

When the system is in a steady state, the system was made to generate a velocity disturbance of 0.2 m/min for $v_0$ in the 4ths ($v_0 = \omega_0 R_0$); $v_4$ was made to generate a velocity disturbance of 0.2 m/min in the 8ths ($v_4 = \omega_4 R_4$). Then, $T_{02}$ is made to generate a random disturbance of size $\pm 5$ N in $1\sim 5$ s time, and $T_{04}$ is made to generate a random disturbance of size $\pm 5$ N in $6\sim 10$ s time, and the results of simulation using parameter self-tuning decoupling controller based on improved ADRC for tension system and PID controller are shown in Figure 8.

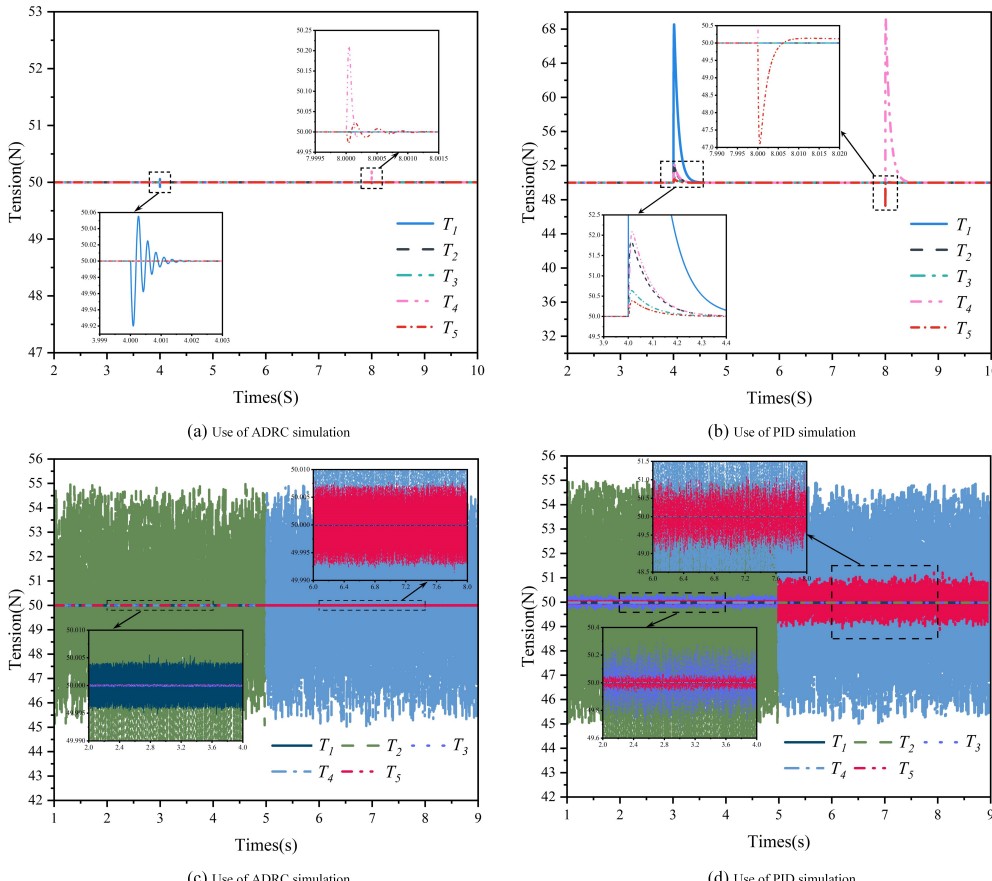

**Figure 8.** Comparison of anti-interference simulation results.

Observing the curves of Group (a) in Figure 8, it is evident that when $v_0$ generates speed disturbance, $T_1$ generates a tension disturbance of 0.16%. Similarly, when $v_4$ generates speed disturbance, $T_4$ and $T_5$ generate tension disturbances of 0.4% and 0.06%, respectively. In the case of Group (b) curves, it can be observed that when $v_0$ generates speed disturbance, $T_1$, $T_2$, $T_3$, $T_4$, and $T_5$ generate tension perturbations of 16%, 3.1%, and 1.2%, and 4%, and 0.6%, respectively. Likewise, when $v_4$ generates velocity disturbance, $T_4$ and $T_5$ generate tension perturbations of 17% and 6%, respectively. The curves of Group (c) suggest that when $T_{02}$ generates sustained tension perturbation, $T_1$ generates a sustained tension perturbation of 0.01%. Similarly, when $T_{04}$ generates sustained tension perturbation, $T_5$ generates a sustained tension perturbation of 0.016%. Lastly, the Group (d) curves suggest that $T_{02}$ produces a sustained tension perturbation, $T_3$ produces a sustained tension perturbation of 0.6%, and $T_5$ produces a sustained tension perturbation of 0.1%. Similarly, $T_{04}$ produces a sustained tension perturbation, and $T_5$ produces a sustained tension perturbation of 3%.

The above results show that under speed disturbance, when using the parameter self-tuning decoupling controller based on improved ADRC for tension system, the maximum relative error of the tension unit is 0.4%; when using the traditional PID controller for tension control, the maximum relative error of the tension unit is 17%; under tension disturbance: when using the parameter self-tuning decoupling controller based on improved ADRC for tension system, the maximum relative error of the tension unit is 0.016%; when using the traditional PID controller for tension control, the maximum relative error of the tension unit is 3%. Compared with the traditional PID control, the parameter self-tuning decoupling controller based on improved ADRC for tension system has better anti-interference ability.

Several researchers have conducted studies on designing decoupled controllers for tension systems. In this paper, the evaluation of the tension control performance of different

controllers is based on the relative tension error. Table 10 presents a comparison of the tension control effect of these controllers.

**Table 10.** Controller performance comparison.

| Controller | Perturbation Mode | Relative Error | Author | Reference |
|:---:|:---:|:---:|:---:|:---:|
| FOPID [1] | Tension disturbance | 4.63% | Meng | [23] |
| REKF-PID [2] | Tension disturbance | 3.74% | Zhang | [24] |
| RD-$H\infty$ [3] | Speed disturbance | 3.86% | Chen | [25] |

[1] Fractional order PID denoted as FOPID. [2] fuzzy PID controller based on REKF denoted as REKF-PID, Robust Extended Kalman filter is REKF. [3] denoted as RD-$H\infty$.

Table 10 shows that under tension disturbance, the maximum relative error of FOPID controller is 4.63%, the REKF-PID controller is 3.74%, and the decoupled controller of tension system designed in this paper is 0.016%; under speed disturbance, the maximum relative error of RD-$H\infty$ controller is 3.86%, and the decoupled controller of tension system designed in this paper is 0.4%. Therefore, the designed decoupling controller in this paper has a better control effect compared with FOPID, REKF-PID, and RD-$H\infty$ controllers under tension disturbance and speed disturbance.

## 5. Conclusions

In this paper, a global coupling model of the tension system of the roll-to-roll coating machine is established, the parameter self-tuning decoupling controller based on improved ADRC for the tension system is designed, and simulation experiments are carried out in Simulink. The experiment utilizes GA to carry out the parameter tuning of ADRC, and the experimental results show that GA can complete the parameter tuning of ADRC well and meet the tension control requirements. Experiments in the decoupling performance simulation: using the decoupling controller designed in this paper, the maximum relative error of the other tension unit is 0.012%, using the traditional PID controller, the maximum relative error of the tension unit is 1.5%, the parameter self-tuning decoupling controller based on improved ADRC for tension system has a better decoupling effect; In the anti-interference simulation: the maximum relative error of the decoupling controller designed in this paper is 0.4% for the tension unit under speed interference and 0.016% under tension interference, compared with 17% and 3% for the conventional PID controller, and the parameter self-tuning decoupling controller based on improved ADRC for tension system has a better anti-interference effect; At the same time, the results of other researchers are analyzed and compared with the decoupling controller designed in this paper, and the results show that the tension system decoupling controller designed in this paper is better in the speed interference and tension interference.

In the future, this paper's decoupled controller will be applied to industrial roll-to-roll coating machines and popularized.

**Author Contributions:** Conceptualization, G.J. and S.L.; methodology, G.J. and S.L.; software, G.J. and K.W.; validation, G.J., S.L. and H.D.; formal analysis, G.J.; investigation, G.J. and C.W.; resources, S.L.; data curation, G.J. and K.W.; writing—original draft preparation, G.J.; writing—review and editing, G.J. and H.D.; visualization, G.J.; supervision, S.L.; project administration, S.L. and K.W.; funding acquisition, S.L. All authors have read and agreed to the published version of the manuscript.

**Funding:** This research was funded by the Key Research and Development Program of Shaanxi Province grant number 2023-YBGY-329, Scientific Research Program Funded by Shaanxi Provincial Education Department grant number 22JY048, Key Research and Development Program of Weinan City grant number ZDYFJH-45 and Lab of Intelligent and Green Flexographic Printing grant number ZBKT202303.

**Institutional Review Board Statement:** Not applicable.

**Informed Consent Statement:** Not applicable.

**Data Availability Statement:** No new data were created or analyzed in this study. Data sharing is not applicable to this article.

**Conflicts of Interest:** The authors declare no conflict of interest. The funders had no role in the design of the study, in the collection, analyses, or interpretation of data, in the writing of the manuscript, or in the decision to publish the results.

## Abbreviations

The following abbreviations are used in this manuscript:

| | |
|---|---|
| ADRC | Active Disturbance Rejection Controller |
| PID | Proportional-Integral-Differential |
| GA | Genetic algorithm |
| TD | The most rapid tracking differentiator |
| NLESF | The first-order nonlinear error feedback control |
| ESO | The first-order dilated state observer |
| FOPID | Fractional Order PID |
| REKF-PID | Fuzzy PID controller based on REKF, Robust Extended Kalman filter is REKF |
| RD-$H\infty$ | Robust Decentralized $H\infty$ |

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
