# Peer review of "A Parameter Self-Tuning Decoupling Controller Based on an Improved ADRC for Tension Systems"

_applsci, doi:10.3390/app131911085_

Round 1

Reviewer 1 Report

In this paper, A parameters self-tuning decoupling controller based on improved ADRC for tension system were investigated, This paper’s subject matter is well within the journal topic areas, however there are a number of problems and uncertainties that need the authors’ serious attention, and a significant and majors revision is required before we can assess it again. The following are problem areas:

1.      Introduction it’s insufficient a more detail is required.

2.      This manuscript is mostly a description of the  simulation results, and lack of the depth discussion.

3.      The manuscript is rather an technical note.. Own results are overrated and not discussed on the basis of comparable studies of other authors. 

4.      Thorough discussion of achieved results in light of results of other authors.

5.      Discussion part is too weak,

6.      Please provides a high resolution figures to meet the journal requirement

7.      Conclusion: It is too bulky. Make it concise form possibly with some numerical results.

8.      Conclusions must be comprehensive and not written like a report.

9.      Reference part is not enough, please added some references from 3  last years

Minor editing of English language required

Reviewer 2 Report

The paper “A parameters self-tuning decoupling controller based on improved ADRC for tension system” designs a parameters self-tuning decoupling controller based on improved ADRC (Active disturbance rejection controller) for tension system. The study is well presented and shows interesting results, so it would be advisable to publish it after changes. Some suggestions:

·        A graphical abstract would add interest to catch the eye

·        In the abstract must introduce novelty of the paper.

·        Please introduce acronyms when first appear in the text of the manuscript

·        Poor quality of the pdf with unexpected (“De, fSihnainthiuoinsL/iulo1g,*o,-Koerlciiadn.gpWdfe)

·        Figures are excellent quality

·        Line 160 what reference is [?] (criterion is satisfied [? ]. The)

·        The conclusions should include valuable results

·        Please could you provide future lines

Reviewer 3 Report

       The current manuscript, “A parameters self-tuning decoupling controller based on improved ADRC for tension system”, designs parameters self-tuning decoupling controller based on improved ADRC for tension system. I recommend undergoing Major corrections, especially for the references, which seem to be missing. My comments are below.

1.      The used template has some issues; the author’s names cannot be seen clearly.

2.      Figure 1: If the figure is not original, please insert a reference in the figure description.

3.      Lines 71 -78: I suggest inserting the signification and description of the terms from the equations in a Table, or separately in an appendix.

4.      Line 92: It seems that you have a missing reference. Please make corrections.

5.      Figure 2: If the figure is not original, please insert a reference in the figure description.

6.      Lines 95-102: Again, I suggest inserting the signification and description of the terms from the equations in a Table, or separately in an appendix.

7.      Line 109: The authors said: “Equation (1) is written in the form of a matrix:” You have a system of equations in (1), not only one equation.

8.      Equation 3: What represent “B(t)”?

9.      Line 139:  It seems that you have a missing reference. Please make corrections.

10.   Line 140: It seems that you have a missing reference. Please make corrections.

11.   Line 141: It seems that you have a missing reference. Please make corrections.

12.   Line 152: It seems that you have a missing reference. Please make corrections.

13.   Line 168: It seems that you have a missing reference. Please make corrections.

14.   Figure 3: If the figure is not original, please insert a reference in the figure description.

Author Response

请参阅附件

Round 2

Reviewer 1 Report

  After Carfuly check, The manuscript has been sufficiently improved to warrant publication in Applied Sciences.  we recommande the publication in it's forme.   

Reviewer 3 Report

The authors have responded to all the comments and suggestions from the reviewer, improving the quality of the manuscript. The manuscript can be published.